# CS12192, a Novel JAK3/JAK1/TBK1 Inhibitor, Synergistically Enhances the Anti-Inflammation Effect of Methotrexate in a Rat Model of Rheumatoid Arthritis

**DOI:** 10.3390/ijms232113394

**Published:** 2022-11-02

**Authors:** Zhengyu Fang, Yiping Hu, Jiajing Dai, Lianhua He, Juan He, Bihua Xu, Xinle Han, Fubo Zhong, Huiyao Lan, Qingwen Wang

**Affiliations:** 1Department of Rheumatism and Immunology, Shenzhen Key Laboratory of Inflammatory and Immunology Diseases, Peking University Shenzhen Hospital, Shenzhen 518036, China; 2Department of Pathology, Peking University Shenzhen Hospital, Shenzhen 518036, China; 3Clinical Research Institute, Shenzhen Peking University—The Hong Kong University of Science and Technology Medical Center, Shenzhen 518036, China; 4Department of Medicine and Therapeutics, Li Ka Shing Institute of Health Sciences, and Lui Che Woo Institute of Innovative Medicine, The Chinese University of Hong Kong, Hong Kong, China

**Keywords:** rheumatoid arthritis, methotrexate, CS12192, collagen-induced arthritis, inflammatory mediators

## Abstract

Rheumatoid arthritis (RA) is a common disease worldwide and is treated commonly with methotrexate (MTX). CS12192 is a novel JAK3 inhibitor discovered by Chipscreen Biosciences for the treatment of autoimmune diseases. In the present study, we examined the therapeutic effect of CS12192 against RA and explored if the combinational therapy of CS12192 and MTX produced a synergistic effect against RA in rat collagen-induced arthritis (CIA). Arthritis was induced in male Sprague-Dawley rats by two intradermal injections of bovine type II collagen (CII) and treated with MTX, CS12192, or the combination of CS12192 and MTX daily for two weeks. Effects of different treatments on arthritis score, X-ray score, pathology, and expression of inflammatory cytokines and biomarkers were examined. We found that treatment with either CS12192 or MTX produced a comparable therapeutic effect on CIA including: (1) significantly lowering the arthritis score, X-ray score, serum levels of rheumatic factor (RF), C-reactive protein (CRP), and anti-nuclear antibodies (ANA); (2) largely alleviating histopathological damage, reducing infiltration of Th17 cells while promoting Treg cells; (3) inhibiting the expression of inflammatory cytokines and chemokines such as IL-1β, TNF-α, IL-6, CCL2, and CXCL1. All these inhibitory effects were further improved by the combinational therapy with MTX and CS12192. Of importance, the combinational treatment also resulted in a marked switching of the Th17 to Treg and the M1 to M2 immune responses in synovial tissues of CIA. Thus, when compared to the monotherapy, the combination treatment with CS12192 and MTX produces a better therapeutic effect against CIA with a greater suppressive effect on T cells and macrophage-mediated joint inflammation.

## 1. Introduction

Rheumatoid arthritis (RA) is a systemic autoimmune disease. Pathologically, chronic inflammation of the synovial membrane, pannus formation, and destruction of articular cartilage and bone are key features, which may result in joint deformity and loss of function. Clinically, RA patients may show fever and fatigue with other systemic manifestations. Autoantibodies, such as RF and anti-cyclic citrulline polypeptide (CCP) antibody, can be found in RA patients’ serum [1,2,3]. However, the pathogenic mechanism of RA remains unclear [4]. 

Disease-modifying antirheumatic drugs (DMARDs) are a class of drugs used for the treatment of RA [5]. The available DMARDs are subdivided into (1) conventional synthetic DMARDs (methotrexate, hydro chloroquine, and sulfadiazine), (2) targeted synthetic DMARDs such as pan-Janus Kinase (JAK)- and *JAK1/2*-inhibitors, and (3) biologic DMARDs such as tumor necrosis factor-alpha (*TNF-α)* inhibitors, *TNF*-receptor (R) inhibitors, *IL-6* inhibitors, *IL-6R* inhibitors, B cell-depleting antibodies, and inhibitors of co-stimulatory molecules [6,7]. Despite the rapid development of new drugs for the treatment of RA, Methotrexate (MTX) remains a common drug used for treating patients with RA clinically [8]. MTX is originally designed as a folate pathway antagonist by inhibiting dihydrofolate reductase (DHFR) when given at very high doses for leukemia (as high as 1 g in a single dose), but it is found that the drug was effective in RA patients when using it at much lower doses (15–25 mg weekly) [9,10]. Mechanistically, MTX increases adenosine signaling to promote the overall anti-inflammatory state [11].

Combining MTX with other drugs for treatment of RA has been investigated, including the combination of MTX with cyclosporine or hydroxychloroquine and sulfasalazine [12,13]. In these studies, the combination therapy can produce a better therapeutic effect against RA than that of the monotherapy, although it remains controversial [14].

CS12192 is a small molecular drug discovered by Shenzhen Chipscreen Biosciences Ltd. that selectively inhibits *JAK3 and* partially inhibits *JAK1* and TANK-binding kinase1 (*TBK1)* [15]. TBK1 is known to play a key role in the induction of interferon signaling during inflammation, which may lead to excessive tissue damage [16,17]. Until now, no *TBK1* inhibitor has been approved for the treatment of RA. In a previous study, CS12192 showed efficacy in multiple autoimmune disease models such as multiple sclerosis, systemic lupus erythematosus, and graft-versus-host disease [15]. Particularly, it shows encouraging therapeutic effects on some autoimmune disease models with a good safety profile [14]. Thus, CS12192 may be a new therapeutic drug for the treatment of autoimmune diseases and autoimmunity-related diseases.

Till now, the therapeutic effect of CS12192 on RA remains unexplored. Therefore, the current study aimed to investigate the therapeutic effect of CS12192 on RA in a rat model of CIA. Treatment with MTX was used as positive control. More importantly, the combination of CS12192 and MTX for treatment on RA was also performed. We found that like MTX, treatment with CS12192 alone produced a comparable therapeutic effect on CIA, which was further improved by the combination therapy of CS12192 and MTX. 

## 2. Results

### 2.1. CS12192, Similar to MTX, Has a Therapeutic Effect on Arthritis Score, Joint Pathological Damage, and Serum Biomarkers in CIA Rats, Which Is Further Improved by the Combined Treatment with CS12192 and MTX 

On day 10 after the secondary induction of CIA, the majority of rats developed CIA as demonstrated by acute inflammation including redness and swelling in the foot and double-stepped joints (Figure 1A). We used joint arthritis scores to measure the severity of joint damage throughout the entire experimental time course and found that the toe and the double-stepped joints were obviously swollen in control-treated CIA rats, which were alleviated in those treated with either CS12192 or MTX (Figure 1A). Strikingly, combined treatment with CS12192 and MTX largely suppressed the arthritis score to the comparable level of normal control rats (Figure 1B). X-ray measurements showed that the joint destruction in the CIA- and CIA+MTX-treated rats was severe, but was significantly improved in those receiving the combined treatment with CS12192 and MTX (Figure 1C). Of importance, CIA rats treated with CS12192 and MTX had much lower X-ray scores when compared with those CIA rats receiving either MTX or CS12192 monotherapy at 28 days after treatment (Figure 1D).

Histopathological, both ankle and knee joint tissues developed severe destruction of the synovial membrane, hyperplasia of the synovial membrane, severe edema and exudation, and a massive infiltration of inflammatory cells in the sub-synovial connective tissue in control-treated CIA rats (Figure 2). All these pathological damages were significantly reduced by treatment with CS12192 or MTX, which were further suppressed in CIA rats receiving the combined CS12192 and MTX treatment (Figure 2). In addition, compared to the CIA rats, monotherapy with either CS12192 or MTX significantly reduced serum levels of RF, CRP, and ANA, which were further inhibited in those treated with CS12192 and MTX (Figure 3). Taken together, these findings demonstrated that CS12192, like MTX, produced a comparable therapeutic effect against RA, which was further improved when the combination treatment was used.

### 2.2. Combination Treatment with CS12192 and MTX produces a Better Inhibitory Effect on Joint Inflammation by Inducing Treg and M2 while Suppressing M1 Immune Responses in Synovial Tissues

We next examined the mechanisms by which the combined CS12192 and MTX treatment produces a better therapeutic effect on RA. Compared to the control, CIA-, CS12192- or MTX-treated groups, the combined treatment with CS12192 and MTX significantly reduced the frequencies of *CD3*+ cells and *CD45*+ cells in rat peripheral blood, although there was little synergistic effect on these changes (Appendix A).

Next, by using two-color immunofluorescence, we clearly detected a marked infiltration of Th17 cells (*CD4*+ *IL17A*+) in the synovial tissues of CIA rats, which was significantly reduced in those receiving either CS12192 or MTX or the combination of CS12192 and MTX therapy (Figure 4A, D). Interestingly, compared to the monotherapy, the combined treatment with CS12192 and MTX largely promoted the Treg cells (*CD4*+ *FOXP3*+) and M2 macrophages (*CD68*+ *CD206*+) while suppressing the *CD68*+ *iNOS*+ M1 macrophages infiltrating the synovial tissues of CIA rats (Figure 4A,C,E).

We further measured the expression levels of proinflammatory cytokines and chemokines in the synovial fluids of the CIA rats by ELISA. Results shown in Figure 5 revealed that the monotherapy with either CS12192 or MTX was able to significantly reduce levels of *TNF-α, IL-1β, CCL2, IL-6, CXCL-1*, and *IL-17A*, which were further significantly reduced in those receiving the combination treatment with CS12192 and MTX (Figure 5A). These findings were further confirmed at the mRNA level as determined by real-time PCR, in which the combination treatment resulted in a much greater inhibition of *TNF-α, IL-1β, CCL2, IL-6, CXCL1* and *IL-17A* expression in the joint tissues (Figure 5B). 

### 2.3. Combination Treatment with CS12192 and MTX Produces a Better Inhibitory Effect on Joint Articular Cartilage Degradation 

Cartilage changes are frequently observed in RA, which are associated with joint destruction. To assess the degradation and destruction of cartilage tissues, safranin O and toluidine blue staining was used to probe the cartilage layers. Large amounts of safranin O staining were observed in the rats of Ctrl group, while rats suffering from CIA exhibited cartilage thinning and erosion in the joints (Figure 6A, B). It was observed that cartilage injury in CIA rats was markedly alleviated when treatment with CS12192 or MTX was received, which was more profound in those given the combination treatment with CS12192 and MTX (Figure 6A, B). The protective effect of CS12192 and MTX on joint damage was further supported at the molecular levels that treatment with combined CS12192 and MTX produced a better inhibitory effect on expression of *COMP, MMP-9* and *IL-1R* when compared to the monotherapy (Figure 6C). 

## 3. Discussion

Increasing evidence shows that the *JAK* signaling pathway plays an important role in various autoimmune-related diseases [18,19]. The *JAK* kinase family includes four subtypes of *JAK*1, 2, 3 and *Tyk*2, which respectively mediate different immune cytokine signal pathways and biological activities [20]. They are closely related to the regulation of the immune system. Thus, targeting the *JAK* pathway has been shown to effectively inhibit autoimmune-related diseases including RA [21]. However, no selective *JAK*3 inhibitor has been developed for the treatment of RA. In the present study, we found that when compared to MTX, treatment with CS12192, a *JAK*3 inhibitor, produced a comparable therapeutic effect on CIA in rats. This was supported by the findings that like MTX, treatment with CS12192 significantly reduced the arthritis score, X-ray score and serum levels of *RF, CRP* and *ANA*, and improved the histopathological damage and joint inflammation, as well as the expression of inflammatory cytokines and chemokines such as *IL-1β, TNF-α, IL-6, CCL2*, and *CXCL1*. Based on these findings, we demonstrated that CS12192 may act as a JAK3 inhibitor capable of inhibiting CIA in rats. Most importantly, we also found that the combined treatment with CS12192 and MTX produced a synergistic effect on CIA. All these findings suggested that the combined therapy with CS12192 and MTX may be a better therapeutic strategy for RA clinically.

The synergistic inhibition of T cells immune responses may be a key mechanism through which the combined treatment with CS12192 and MTX produced a better inhibitory effect on CIA. We found that the combined therapy with CS12192 and MTX significantly induced regulatory T cells while suppressing Th17 immune responses when compared with the monotherapy. It has been well established that Treg cells are protective, whereas Th17 cells are pathogenic in patients and animals with RA [22,23]. Thus, the superior inhibitory effects of combined CS12192 and MTX on CIA over the monotherapy may be associated with the switching of T cell immune responses from Th17 to Treg. Published studies showed that specific JAK/STAT pathways play a critical role in the functional differentiation of distinct Th subsets, and heme oxygenase-1 exerts its inhibitory effect on Th17 cell differentiation by directly blocking STAT3 phosphorylation [24]. In the current study, CS12192 might enhance the therapeutic effect of MTX on RA through the inhibition of JAK3/STAT pathway, which needs to be further elucidated.

The synergistic blockade of M1 macrophages while promoting M2 macrophage immune responses may be another mechanism by which the combined treatment with CS12192 and MTX produced a synergistic suppression on CIA. Both M1 macrophages and M2 macrophages are closely related to inflammatory responses, among which M1 macrophages participate in pro-inflammatory responses, while M2 macrophages are mainly involved in anti-inflammatory responses. Improving the inflammatory environment by modulating the activation state of macrophages is an effective method for the treatment of RA [25]. In the present study, the synergistic effect of the combinational treatment with CS12192 and MTX on RA may be associated with more effective inhibition of M1 macrophage activation, while promoting M2 immune responses when compared with the monotherapy. It has been reported that inhibition of JAK3/STAT3 pathway could modulate the plasticity of M1 and M2 macrophages in RA [26], which is in agreement with our findings that CS12192 also influenced the M1/M2 balance in the present study. 

Monocyte chemoattractant protein-1/*CCL2* (a ligand of *CCR2*) can attract monocytes, T cells, natural killer cells and basophils [27]. *CCL2* is highly expressed in synovial tissue and synovial fluid in RA patients, and synovial macrophages are the dominant source of *CCL2* [28,29]. The levels of *CCL2* correlate significantly with the levels of IL-1β, IL-6 and CXCL8 in culture supernatants of synovium from RA patients [30]. In a rodent model, CXCL1 has been reported to mediate neutrophil recruitment to the joints of arthritic mice, and its upregulation in ankle and synovial fluid parallels disease progression [31,32]. Although the mechanisms whereby the combination therapy with CS12192 and MTX produced a better therapeutic effect on CIA are largely unclear, it may be associated with the differential effect of CS12192 versus MTX on macrophage activation during CIA, as treatment with CS12192 produced a relatively better inhibitory effect on the expression of *CCL2*, *IL*-6 and *CXCL*1 when compared to that of MTX, although MTX produced more favorable inhibition of TNF-α expression than that of CS12192. 

In addition, we also found that the combination treatment of CS12192 and MTX had a synergistic effect on preventing cartilage destruction in the CIA rats. It is well known that *COMP*, a prominent constituent of articular cartilage, has been reported to increase in patients with knee osteoarthritis (OA) and early RA [33,34]. Importantly, MMP family members are also the major enzymes that degrade the components of the extracellular matrix [35]. Under pathologic conditions, the levels of *MMP* expression increase considerably, resulting in aberrant connective tissue destruction. The activity of *MMP*s is regulated by their endogenous inhibitors, tissue inhibitors of metalloproteinases (*TIMPs*) [36]. *TIMPs* regulate the breakdown of extracellular matrix components and play an important role in tissue remodeling and growth, in both physiological and pathological conditions [37]. In the present study, we found that the combinational treatment with CS12192 and MTX produced a synergistic effect on the inhibition of *COMP* and *MMPs* while increasing *TIMP*, resulting in a better protective effect on joint damage.

Since CS12192 was a gift from Dr. Zhou, we also compared our work on CIA rats to the results reported by their team. They found that CS12192 significantly alleviated disease scores, hind-paw swelling, bone erosion and joint tissue inflammation of rat CIA model in a dose-dependent manner [15]. In the present study, when CS121912 was used alone on CIA rats, similar results were also observed. CS12192 could reduce joint inflammation and cartilage damage, which has a similar effect with MTX.

MTX has also been shown to exert *JAK* inhibition of *JAK*2 and *JAK*1 [38]. In the present study, when it was used together with the new *JAK*3 and *TBK*1 inhibitor, there was a synergistic effect, which was a novel finding. It has been found that the combination of MTX with Leflunomide, or *JAK*1/*JAK*3 inhibitors, leads to better clinical outcomes than monotherapy, while the combination with *JAK*1/*JAK*2 or *JAK*1 specific inhibitors does not seem to exert additive clinical benefit [39]. This may also partially explain the result that the combination of CS12912 and MTX has better therapeutic effect in CIA rats.

## 4. Materials and Methods 

### 4.1. Collagen-Induced Arthritis Model in Rats

A total of 45 Sprague-Dawley (SD) rats (male, 8 weeks, 180–200 g) were purchased from Guangdong Medical Laboratory Animal Center. Rats had access to food and water ad libitum and were housed in an environmentally controlled room with 12 h light/dark periods. All rats were free to move in the cage. 

Arthritis was induced with 40 rats by bovine collagen type II (Chondrex; dissolved in 0.05 M acetic acid) and IFA (incomplete Freund’s adjuvant, Chondrex) as previously described [40]. In short, CII–IFA emulsified liquid was mixed by bovine collagen type II and incomplete Freund’s adjuvant at the same volume and this emulsified mixture was injected to the dorsum of rats’ tails, 2 cm distal from the base with a volume of 200 μL for each rat. Following the first immunization, the second immunization as a booster was performed a week after the first immunization with 100μL of CII-IFA for each rat. 

CS12192 (structure information see Appendix A) was a kind gift from Dr. Zhou You of CHIPSCREEN BIOSCIENCES Corporation in Shenzhen, China. For in vivo administration, CS12192 compound was suspended in pure water for oral gavage. Treatment drugs were started on the day of secondary immunization; CIA rats received an oral drug treatment with CS12192, MTX, or the combination of CS12192 and MTX daily as shown in Table 1 until day 28. CIA group rats receiving phosphate-buffered saline (PBS) treatment were used as a control and a group of healthy untreated rats was also used as normal control. After 28 days of treatment, the rats were sacrificed. Serum levels of rheumatoid factor (RF), C-reactive protein (CRP) and antinuclear antibody (ANA) in serum, synovial fluid and collected tissues were detected by ELISA.

### 4.2. Assessment of Joint Arthritis

The joint arthritis scores were evaluated at 0, 7, 14, 21, 28 days after the second immunization. The four paws of the rats were quantitatively scored and the total arthritis scores of each group were calculated according to the joint arthritis score of each rat [40,41]: 0 = no redness; 1 = red spot s or mild swelling; 2 = moderate swelling of the joints; 3 = severe swelling; 4 = joint deformation without the ability to bear weight. The total score for arthritis is 16 points. 

### 4.3. X-ray Examination

The radiological test (35 KV, 20 s; Faxitron, X-ray) was completed to evaluate the treatment effects of each group of rats at day 28. Then, the semi-quantitative integral analysis and its criteria are as follows [42,43]: 0 = a normal joint structure and clearance; 1 = joint space is suspected to be narrowed along with slight erosion of the joint surface; 2 = joint space and articular surface significantly narrowed and eroded; 3 = blurred joint space, partial erosion of subchondral bone and mild deformity of the articular surface; 4 = cannot distinguish the joint space, the cartilage of the subchondral bone and joint structural obvious abnormalities; 5 = the whole joint is indistinguishable and severely damaged, accompanied by physical disability. All semi-quantitative integrals come from two different experiments, and the results are the average of them.

### 4.4. Histopathological Examination

Upon sacrifice on day 28, the ankles were removed, fixed in 10% formalin for 24 h, decalcified in formic acid for 30 days, and then embedded in paraffin, sectioned, and stained with hematoxylin and eosin (H&E) or safranin O [44]. The H&E-stained sections were scored for inflammation. The following scale was used [45]: 0 = no inflammation; 1 = synovial inflammation, mild infiltration; 2 = mild synovial inflammation; 3 = moderate synovial inflammation; and 4 = synovium highly infiltrated with many inflammatory cells.

Safranin O staining was scored for cartilage damage with the following scale: 0, no destruction; 1, slight reduction; 2, moderate reduction; 3, severe reduction; and 4, absence of staining. The two hind paws of the rats were quantitatively scored and the total scores of each group were calculated and analyzed among the groups.

### 4.5. Immunofluorescent Staining

For immunofluorescent staining, synovial tissues obtained from rats in different groups were fixed in formalin, and 4 µm sections were cut for two-color immunofluorescent staining. The sections were fixed in 4% paraformaldehyde (pH 7.4) for 10 min at 37 °C and permeabilized by 0.1% Triton X-100. Then, the sections were cultured with monoclonal antibodies against rat *CD4* (Catalog NO. ab237722, Abcam, Waltham, MA, USA) together with *FOXP3* (Catalog NO. ab75763, Abcam, USA) or *IL-17A* (Catalog NO. ab214588, Abcam, USA), *CD68* (Catalog NO. ab125212, Abcam, USA) together with *iNOS* (Catalog NO. NB300-605SS, Novus Biologicals, Centennial, CO, USA) or *CD206* (Catalog NO. 24595T, Cell Signaling Technology, Danvers, MA, USA) overnight at 4 °C. The sections were incubated with secondary antibody for 1h at room temperature, and then labeled with fluorescent dye (FITC or CY3) secondary antibody with accompanying nucleus (DAPI) for dark incubation at room temperature for 20 min. After washing, the slides were air-dried and fixed with patch media (including anti-fading agent) and examined under an Olympus bioluminescent microscope (IX2-ILL100). 

### 4.6. Measurement of Serum Rheumatoid Markers and Pro-Inflammatory Cytokines in Joint Fluids by ELISA 

Serum RF was detected by a Rat RF ELISA Kit (Catalog Number CSB-E13666r, CUSABIO, Houston, TX, USA), and serum levels of CRP and ANA were assayed using cytokine ELISA kits (Invitrogen, Waltham, MA, USA), following the manufacturer’s protocol. 

The levels of TNF-α, IL-6, IL-17a (Life Technologies, Carlsbad, CA, USA), CCL2, IL-1β (Biolegend, San Diego, CA, USA) and CXCL-1 (Abcam, Cambridge, UK) in synovial fluids were measured by ELISA using commercially available kits according to the instruction of the manufacturers. In order to obtain the synovial fluid, the right knee of each rat was cut and the joint cavity was exposed under aseptic conditions on day 28. The cavity was then subjected to lavage with 1 mL saline and 0.5 mL synovial fluid was aspirated. The synovial fluid specimens were centrifuged at 4500 r/min for 10 min, then the supernatant was stored in Eppendorf tubes at −80 °C. And the detection and calculation service of ELISA were provided by Shanghai Enzyme-linked Biotechnology Co., Ltd., Shanghai, China. 

### 4.7. RNA Extraction and Real-Time PCR Analysis

We used total RNA extraction kit (no. B511311-0500, Shanghai Sangon Biotech, Shanghai, China) to extract RNA from synovial tissues of the control and CIA rats according to the instructions, and purity of the RNA was tested. RNA was reverse-transcribed using a reverse transcription kit (Catlog#K1622, Fermantas, Waltham, MA, USA). Quantitative PCR was performed in a Bio-Rad Chromo4 real-time PCR system (Bio-Rad, Hercules, CA, USA) with primers listed in Table 2. The mRNA levels of *IL-1β, TNF-α, CCL2, IL-6, IL-10, CXCL-1, COMP, MMP9*, *Timp1* and *IL1R* were expressed as the ratio relative to the β-actin mRNA in each sample.

### 4.8. Statistical Analysis

Significant differences between treatments were determined using One-way ANOVA, the Kruskal–Wallis test and Dunn’s multiple comparisons. All statistical analyses were performed using GraphPad Prism V5.0 (GraphPad Software Inc., San Diego, CA, USA). *p*-values less than 0.05 were considered significant.

## 5. Conclusions

In summary, the results of the current study demonstrate that like MTX, CS12192 can function as a *JAK3* inhibitor capable of inhibiting CIA in rats. Importantly, compared to the monotherapy, the combinational therapy using MTX and CS12192 can produce a synergistic effect on CIA by inhibiting joint inflammation, synovial cellularity, expression of pro-inflammatory cytokines, and bone destruction in a CIA rat model. Of importance, the switching of the Th17 to Treg and M1 to M2 immune responses may be key mechanisms through which the combinational treatment with CS12192 and MTX produces a better therapeutic effect against CIA, although the synergistic mechanisms remain largely unclear. 

## Figures and Tables

**Figure 1 ijms-23-13394-f001:**
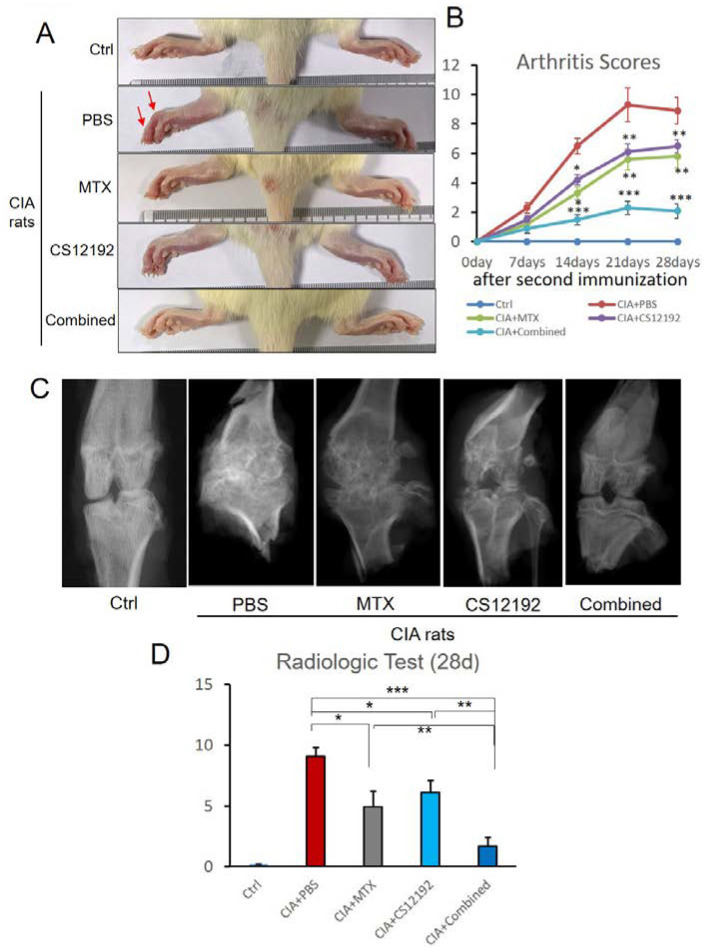
**Effects of MTX (Methotrexate), CS12192, or the combination of CS12192 and MTX on the severity of CIA rats:** (**A**) The representative hind paws of CIA rats from different experimental groups (the red arrows are red and swollen joints). (**B**) The arthritis scores of different groups were quantified throughout the experimental period; medians were compared using One-way ANOVA; * *p* < 0.05; ** *p* < 0.01; and *** *p* < 0.001. (**C**) The results of X-ray examination. (**D**) X-ray semiquantitative analysis results; each bar represents the mean ± SEM for 10 CIA rats in different groups; * *p* < 0.05; ** *p* < 0.01; and *** *p* < 0.001 as indicated.

**Figure 2 ijms-23-13394-f002:**
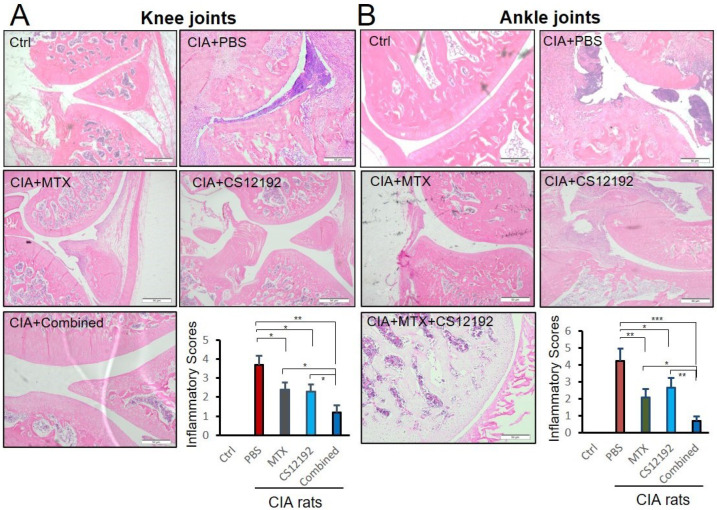
**Therapeutic effects of MTX, CS12192, or the combination of CS12192 and MTX on joint histopathology of CIA rats.** (**A**) knee and (**B**) ankle joint tissue sections of CIA rats in different groups by Hematoxylin and Eosin staining. Scale bar = 50 µm. Inflammation scores were calculated as described in the Methods. Data represent the mean ± SEM for groups of 10 rats from different groups. * *p* < 0.05; ** *p* < 0.01; and *** *p* < 0.001 as indicated.

**Figure 3 ijms-23-13394-f003:**
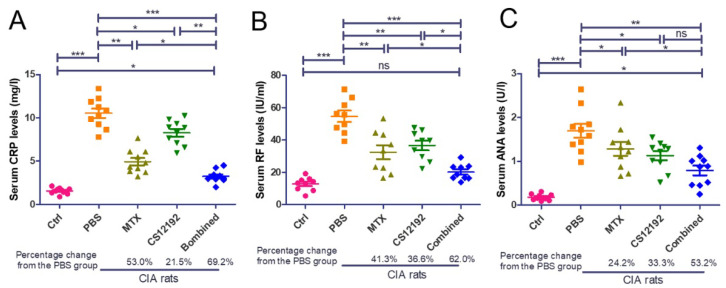
Therapeutic effects of MTX, CS12192 or the combination of CS12192 and MTX treatments on serum levels of CRP, RF, and ANA in CIA rats. Serum levels of CRP, RF, and ANA in CIA rats were measured by ELISA on day 28 after the first immunization. (**A**) Serum levels of CRP. (**B**) Serum levels of rheumatic factor (RF). (**C**) Serum levels of anti-nuclear antibody (ANA). Each dot represents one rat. * *p* < 0.05; ** *p* < 0.01; and *** *p* < 0.001 as indicated.

**Figure 4 ijms-23-13394-f004:**
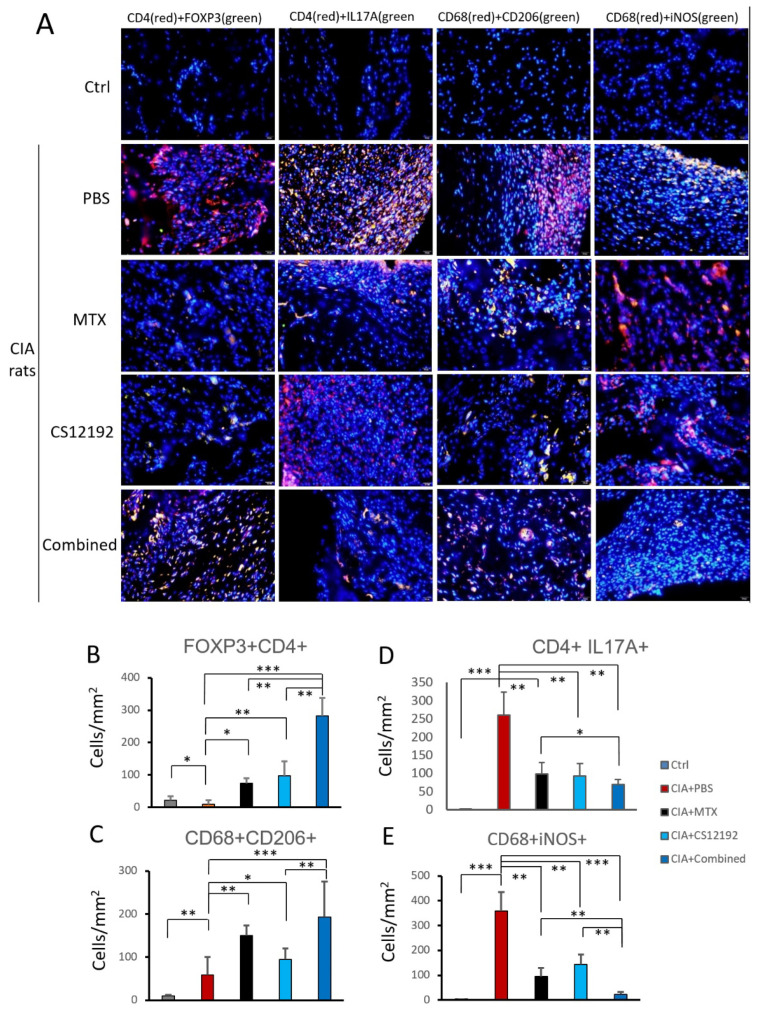
Effects of MTX, CS12192, or the combination of CS12192 and MTX treatments on T cell and macrophage infiltration and activation in CIA rats on day 28 by two-color immunofluorescence, (**A**) Representative two-color immunofluorescence for Treg cells (CD4+ FOXP3+), Th17 cells (CD4+ IL-17A+), M2 macrophages (CD68+ CD206+), and M1 macrophages (CD68+ iNOS+) infiltrating joint tissues on day 28 after the first immunization of CIA rats receiving different treatments. Nuclei were stained with DAPI (blue); (**B**–**E**) Quantitative analysis of T cell and macrophage subsets infiltrating the joint tissues. Each bar represents the mean ± SEM for 10 CIA rats in different groups. Scale bar = 50 μm. * *p* < 0.05; ** *p* < 0.01; and *** *p* < 0.001 vs. CIA group.

**Figure 5 ijms-23-13394-f005:**
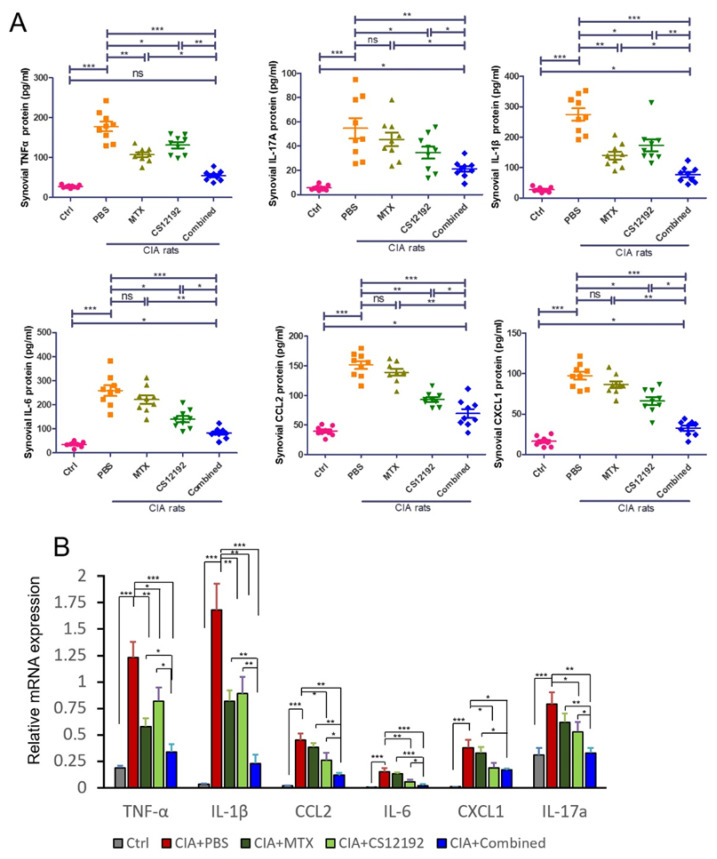
Therapeutic effects of MTX, CS12192, or the combination of CS12192 and MTX treatments on expression of pro-inflammatory cytokines in synovial fluids by ELISA and joint tissues by real-time PCR on day 28. (**A**) ELISA measurements of TNF-α, IL-1β, CCL-2, IL-6, CXCL-1, IL-17A levels in synovial fluids in CIA rats receiving different treatments. Each dot represents one rat. (**B**) Real-time PCR detects mRNA expression of TNF-α, IL-1β, CCL-2, IL-6, CXCL-1, IL-17A in the synovial tissues of CIA rats. Each bar represents the mean ± SEM from 10 animals per group * *p* < 0.05; ** *p* < 0.01; and *** *p* < 0.001 as indicated.

**Figure 6 ijms-23-13394-f006:**
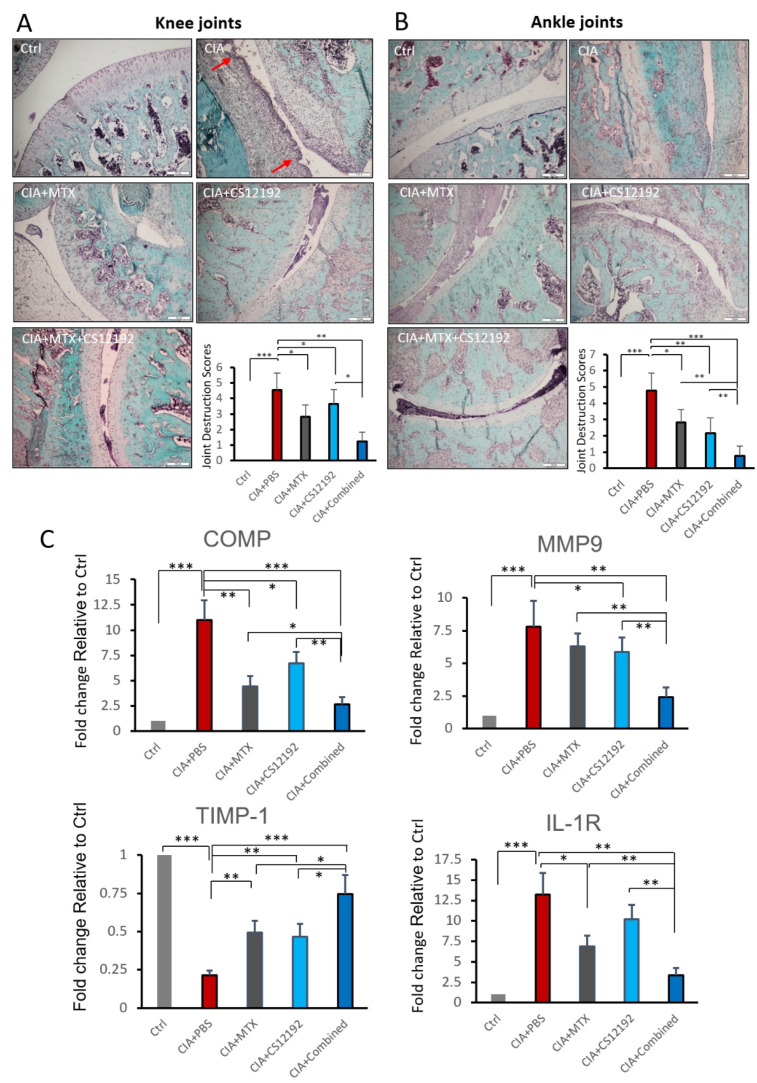
Effects of MTX, CS12192, or the combination of CS12192 and MTX treatments on cartilage destruction determined by safranin O-fast green staining and real-time PCR in joint tissues of CIA rats on day 28. (**A**,**B**) The safranin O-fast green staining joint tissue sections of knee (the red arrows are cartilage destruction areas) (**A**) and ankle (**B**). The cartilage destruction scores were measured as described in the Methods and are presented as the mean ± SEM from groups of 10 rats. Scale bar, 50 µM. (**C**) Real-time PCR analysis for relative mRNA expression of COMP, MMP-9, TIMP-1 and IL-1R in synovial tissues of CIA rats in response to different treatments. Results are presented as the mean ±SEM from groups of 10 rats. * *p* < 0.05; ** *p* < 0.01; and *** *p* < 0.001 as indicated.

**Table 1 ijms-23-13394-t001:** Experimental Design of the current study.

Group Name and Number of Rats	Day 0 (First Immunization)	Day 7 (Second Immunization and Stat Point of Treatment)	Day 35
Untreated Group (5)	Normal rats	Normal rats	End of Experiments
CIA * group (10)	CII-IFA *	CII-IFA+PBS * (1 ml/qd *)	
MTX * group (10)	CII-IFA	CII-IFA +MTX (0.2 mg/kg/qd)	
CS12192 group (10)	CII-IFA	CII-IFA +CS12192 (40 mg/kg/bid *)	
MTX + CS1219 group (10)	CII-IFA	CII-IFA +MTX (0.2 mg/kg/qd) + CS12192 (40 mg/kg/qd)	

* Abbreviation: collagen-induced arthritis, CIA; type II collagen in complete Freund’s adjuvant, CII-IFA; methotrexate, MTX; phosphate-buffered saline, PBS; quaque die, qd; bis in die, bid.

**Table 2 ijms-23-13394-t002:** Sequences of primer sets for real-time PCR analysis.

Gene (Rat)	Forward (5′-3′)	Reverse (5′-3′)
TNF-α	GTCGTAGCAAACCACCAAGC	CCACCAGTTGGTTGTCTTTGA
IL-1β	CTGCCAAGTCAGGTCTCTCA	AGGTAAGTGGTTGCCTGTCA
CCL2	CCTGCTGCTACTCATTCACTG	ACAGCTTCTTTGGGACACCT (94)
IL-6	AGTGGCTAAGGACCAAGACC	ACCACAGTGAGGAATGTCCA (136)
CXCL1	CTCCAGCCACACTCCAACAGA	CACCCTAACACAAAACACGAT
IL-17a	GAAGGCCCTCAGACTACCTC	GTGCCTCCCAGATCACAGAA (96)
Comp	GGGATGGACACCAAGACTCC	TCATCACAGGCATCACCCTT (97)
MMP9	CCCTACTGCTGGTCCTTCTG	CTTCCAATACCGACCGTCCT(36)
IL-1R	TTCAGGGCACACATGTCCTA	ATGATCTGGTGGCAGTCACA (126)
Timp-1	CATGGAGAGCCTCTGTGGAT	TATGCCAGGGAACCAGGAAG (125)
β-actin	CAACGGCTCCGGCATGTGC	CTCTTGCTCTGGGCCTCG

## Data Availability

The original contributions presented in the study are included in the article/Appendix A. Further inquiries can be directed to the corresponding authors.

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
