# Peer review of "CS12192, a Novel JAK3/JAK1/TBK1 Inhibitor, Synergistically Enhances the Anti-Inflammation Effect of Methotrexate in a Rat Model of Rheumatoid Arthritis"

_ijms, 2022, doi:10.3390/ijms232113394_

Round 1
Reviewer 1 Report
Fang, et al. studied the anti-inflammatory effect of CS12192, alone and in the combination with MTX on rat CIA models. Th17/Treg cells and M1/M2 macrophages were involved.
Comments:
1.The therapeutic effect of CS12192, however, had been partially proved in rat CIA models in a previous report (PMID: 31634789).
2.The chemical structure of CS12192 needs to be shown.
3.Did the authors evaluate the side effect of this medication?
4. I noticed the comparisons between CIA+MTX and CIA+combination groups were lack in some experiments. This is important to support an additional benefit of CS12192 to the background treatment of MTX.
5.Which joints were shown in Figure 1C, knees or ankles? Arthritis score should be evaluated throughout the experiment, not only on day 28.
6. It seems MTX did not suppress IL-17a, IL-6, CCL2 and CXCL1 in synovial fluid in Fig 5A.
Reviewer 2 Report
Fang et al. reported that CS12192 could synergistically achieve anti-inflammation effects with MTX. The data is clearly presented. However, the molecular mechanisms regarding of Treg/Th17 and M1/M2 conversion by the conditional therapy needs to be explored to some degree.
Reviewer 3 Report
Summary: The manuscript investigates the therapeutic effect of a selective JAK3 inhibitor on a rat model of rheumatoid arthritis (RA) and compares the effects of the new compound alone to a combination therapy with methotrexate (MTX). The study demonstrates novelty by exploring the potential therapeutic effect of a selective JAK3 inhibitor in RA and answers a relevant question by comparing the effect to first-line RA therapeutic MTX.
General comments
The manuscript represents a great scientific value by addressing a clinically relevant question in the field of rheumatoid arthritis. The hypothesis is clear, and the methods and statistical tests chosen are appropriate. The results are presented clearly and assist an easy interpretation for the reader. The conclusions are in agreement with the data presented. Overall, the manuscript is well-organized, however, it would benefit from a close editing due to multiple stylistic errors breaking the flow of the narrative.
Specific comments
Introduction:
1. define JAK and TBK1 at first appearance
Methods and results:
1. Please, justify the dose selection for CS12192. Shan, S et al. 2019 used 20, 40 and 80 mg/kg in a rat CIA model; how was 40 mg/kg selected for the present study?
2. Please, indicate the vehicle for CS12192
3. Section 4.1. “after 4 weeks of treatment”. Please, clarify the duration of drug treatment. Drug treatment was initiated on Day 10 and continued until Day 28.
4. Table 1. Please, define qd and bid abbreviations
5. Section 4.2. the methods imply that quantitative scoring was performed only on Day 28. What is the reason behind that? In my opinion, scoring at multiple time points would deliver more information. If one day, justify why Day 28 was chosen as swelling and redness tends to resolve after the acute phase without treatment in some arthritis models
6. Section 4.4 redundant text, unclear what belongs to section 4.5
7. Section 4.7 indicate what tissues or cells were used to extract RNA
8. Fig 1D. Indicate if data represent mean and SEM
Discussion and conclusion:
1. The terms synergistic and additive are used interchangeably throughout the manuscript. Please, use the terms describing the effect of the combined therapy more precisely and consistently, as a synergistic effect implies a greater effect than additive.
2. Conclusion, first sentence: please, revise. The data do not demonstrate that CS12192 is a selective JAK3 inhibitor.
3. Please, pay close attention to typos and stylistic errors throughout the manuscript.
Round 2
Reviewer 1 Report
The authors have replied all my questions.